# Thermal Performance Analysis of a Nonlinear Couple Stress Ternary Hybrid Nanofluid in a Channel: A Fractal–Fractional Approach

**DOI:** 10.3390/nano14221855

**Published:** 2024-11-20

**Authors:** Saqib Murtaza, Nidhal Becheikh, Ata Ur Rahman, Aceng Sambas, Chemseddine Maatki, Lioua Kolsi, Zubair Ahmad

**Affiliations:** 1Department of Mathematics, Faculty of Science, King Mongkut’s University of Technology Thonburi (KMUTT), 126 Pracha Uthit Rd, Bang Mod, Thung Khru, Bangkok 10140, Thailand; saqib.murtaza@mail.kmutt.ac.th; 2Department of Chemical and Materials Engineering, College of Engineering, Northern Border University, Arar P.O. Box 1321, Saudi Arabia; nidhalmohamedkamel@gmail.com; 3Department of Mathematics, City University of Science and Information Technology, Peshawar 25000, Pakistan; ataurrahman.at80@gmail.com; 4Faculty of Informatics and Computing, Universiti Sultan Zainal Abidin, Besut Campus 22200, Terengganu, Malaysia; acengsambas@unisza.edu.my; 5Department of Mechanical Engineering, Universitas Muhammadiyah Tasikmalaya, Tasikmalaya 46196, Indonesia; 6Department of Mechanical Engineering, College of Engineering, Imam Mohammad Ibn Saud Islamic University (IMSIU), Riyadh 11432, Saudi Arabia; casmaatki@imamu.edu.sa; 7Department of Mechanical Engineering, College of Engineering, University of Ha’il, Ha’il City 81451, Saudi Arabia; lioua_enim@yahoo.fr; 8Department of Mathematics and Physics, University of Campania “Luigi Vanvitelli”, 81100 Caserta, Italy

**Keywords:** ternary nanofluid, couple stress fluid, viscous dissipation, fractal fractional derivative, Crank–Nicolson scheme

## Abstract

Nanofluids have improved thermophysical properties compared to conventional fluids, which makes them promising successors in fluid technology. The use of nanofluids enables optimal thermal efficiency to be achieved by introducing a minimal concentration of nanoparticles that are stably suspended in conventional fluids. The use of nanofluids in technology and industry is steadily increasing due to their effective implementation. The improved thermophysical properties of nanofluids have a significant impact on their effectiveness in convection phenomena. The technology is not yet complete at this point; binary and ternary nanofluids are currently being used to improve the performance of conventional fluids. Therefore, this work aims to theoretically investigate the ternary nanofluid flow of a couple stress fluid in a vertical channel. A homogeneous suspension of alumina, cuprous oxide, and titania nanoparticles is formed by dispersing trihybridized nanoparticles in a base fluid (water). The effects of pressure gradient and viscous dissipation are also considered in the analysis. The classical ternary nanofluid model with couple stress was generalized using the fractal–fractional derivative (FFD) operator. The Crank–Nicolson technique helped to discretize the generalized model, which was then solved using computer tools. To investigate the properties of the fluid flow and the distribution of thermal energy in the fluid, numerical methods were used to calculate the solution, which was then plotted as a function of various physical factors. The graphical results show that at a volume fraction of 0.04 (corresponding to 4% of the base fluid), the heat transfer rate of the ternary nanofluid flow increases significantly compared to the binary and unary nanofluid flows.

## 1. Introduction

The two mathematical ideas that have revolutionized the world are differentiation and integration. For the layperson, these two mathematical tools have two straightforward applications: acceleration and velocity. In addition to these, we might also describe the surface and volume using the concept of integration. The field of differential and integral equations represents the most significant uses of these mathematical skills. In the existing literature, two classes of differential operators exist, and each of the classes has their subclasses. In the first stage, the differential operators are developed based on the rate of change, and these operators are classical, conformable, and fractal derivatives. Of these operators, the classical derivative was discussed by two mathematicians: Leibniz and Newton. Recent research has revealed conformability and fractality to be extremely significant mathematical tools that can be applied to handle local setting-related issues. More precisely, these are the problems that cannot be resolved with the classical derivative. These novel differential operators could be used to model a wide variety of physical issues that occur in nature. Newton’s calculus works only for a certain group of problems that contain continuous space and boundaries. In classical calculus, the phenomena that contain unsmooth boundaries (porous structures) and discontinuous space can be modeled and examined using ordinary derivative operators. In classical mechanics, the theory of continuous fluid fails to account for the influence of molecular size on fluid turbulence. That’s why, to analyze the fluid on a molecule scale, we need to establish an appropriate model for such investigations. However, fractional derivative operators are growing rapidly and are crucial in many scientific and technical disciplines. Fractional derivatives are used to describe those phenomena that cannot be handled by non-fractional calculus. Fractional differential equations are widely used to model a variety of everyday physical problems that contain memory effects. A novel approach to differentiation has been put out recently, in which the operator includes fractional order derivatives and fractal dimensions. It is relatively new to study the concept of differential and integral operators, and not many studies have been conducted on it. It was shown by Abdon Atangana in [1] that fractal calculus and fractional calculus are related, as well as fractal–fractional differential and integral operators. In addition, several unique features and numerical approximations of the operators were showcased, which may prove beneficial in resolving practical issues.

The low thermal conductivity of many fluids contributes to low thermal performance in energy conservation systems. This problem can be overcome by suspending solid nanoparticles in fluids to improve thermal performance. A new class of heat transfer fluids known as nanofluids has been developed by stably suspending nanoparticles in conventional heat transfer fluids. The term “nanofluid,” originally used in 1995 by Choi [2], refers to a fluid that contains nanometer-sized particles (typically ranging from 1 to 100 nanometers in diameter) dispersed within a base fluid, such as water, oil, or ethylene glycol. These nanoparticles are designed to enhance the thermal, rheological, or transport properties of the base fluid, enabling improved heat transfer and flow characteristics for various engineering and industrial applications. Nanoparticle use is common in biological and industrial applications [3,4]. Nanoparticles have great potential in the energy sector. The usage of nanoparticles can be advantageous in the fields of energy conversion, energy storage, and energy conservation. Other related analyses can be found in [5,6,7,8,9,10,11]. Different experts assess the effects of different frameworks on the forced and natural convection of nanofluid flows [12,13]. Thermodynamic performance can be improved by nanofluids, which is important for industries, especially those that employ solar thermal systems for heating or cooling. Due to their remarkable wettability and dispersion properties, nanofluids may also be employed in complicated fluid engineering to create major nanostructured materials and clean surfaces [14]. The delivery of nanomedicine [15] is another application of nanofluids. The strong thermal conductivity of nanofluids is essential for reducing clogging in transport medium walls, achieving high energy productivity, improving performance, and reducing costs [16]. Nanofluids, which involve the dispersion of single nanoparticles in base fluids, have significantly improved heat transfer performance compared to conventional fluids. However, when compared to hybrid nanofluids—fluids with multiple types of nanoparticles mixed—they have several limitations. First, nanofluids with only one type of nanoparticle offer limited thermal conductivity enhancement, whereas hybrid nanofluids leverage the complementary properties of different nanoparticles to achieve a greater heat transfer rate. Additionally, hybrid nanofluids allow for better tuning of physical properties, such as thermal conductivity and viscosity, by varying the composition and ratio of the different nanoparticles, which are less flexible with single component nanofluids. “Hybrid nanofluids” are utilized to achieve the necessary thermal characteristics. Makishima [17] asserts that mixing two or more distinct types of nanoparticles into a single base fluid can result in the formation of a hybrid nanofluid. The “hybrid nanofluid”, a cutting-edge kind of nanofluid used in many engineering problems, heat turbines, heat generators, and cooling systems, showed a potential increase in heat transfer rate. Sarkar et al. [18] found that the thermal performance and pressure drop of the hybrid nanofluid is better than that of the unary nanofluid. Waini et al. [19] investigated the influence of transpiration on the flow of the hybrid nanofluid over expanded metal sheets. The authors suspended *Al_2_O_3_* and Cu nanoparticles in water. They found that the hybrid nanofluid flow performed better compared to the unary fluid flow. Due to their improved thermophysical properties, nanofluids are more efficient in convection processes. Nanoparticles can be suspended in a base fluid in a variety of ways. To improve the efficiency of heat transfer in the scientific field, researchers have recently used three different types of nanoparticles in a base fluid. The term ternary nanofluid refers to such a homogeneous suspension in which three different nanoparticles are dispersed in a base fluid. Manjunatha et al. [20] examined the ternary nanofluid flow over a stretching sheet. A homogenous and stable mixture of ternary nanofluid has been developed by the uniform dispersion of *Al_2_O_3_*, *SiO_2_*, and *TiO_2_* in water. They analyzed the flow equations using *RK-4* techniques and found that ternary nanofluid shows advanced thermal performance compared to hybrid and unary nanofluid flow. Ahmad et al. [21] analyzed the ternary nanofluid in a square flow conduit. In their study, they developed a ternary nanofluid mixture by dispersing *ZnO*, *Al_2_O_3_*, and *TiO_2_* in distilled water. Recent research on ternary hybrid nanofluids can be found in [22,23,24,25].

One important factor in improving industrial processes and energy efficiency is the modern materials’ thermal performance. Within this framework, the investigation of ternary nanofluids with nonlinear couple stress in a channel signifies a noteworthy advancement toward inventive approaches to thermal regulation. Although there is a vast amount of literature on the subject, the fractal–fractional model of ternary nanofluid flow has not yet been explored. To fill this gap in the literature, the authors assumed a ternary nanofluid model for couple-stress fluid flow with nonlinear couple-stress behavior. The mathematical model was formulated in terms of classical order nonlinear PDEs and then fitted with a generalized FFD operator. The final solution was obtained using the Crank-Nicolson scheme. In addition to advancing fundamental knowledge, our study tries to offer useful insights that will propel the advancement of energy-efficient technology in the future.

## 2. Description of the Problem

An analysis has been conducted on a dynamic model of a nonlinear coupling stress trihybrid nanofluid in a channel with a length of l. The study also takes into account the combined effect of heat radiation and viscous dissipation. Initially, during the study, it is assumed that both the channel and the fluid are in a state of rest, with a thermal field denoted as ${Τ}_{s}$. However, by time τ>0, the right plate has been perturbed with a velocity of magnitude $U_{0}H\left(\tau \right)$. This disturbance is then transmitted to the fluid, causing it to initiate motion. Simultaneously, the fluid’s temperature rises ${Τ}_{s}+\left({Τ}_{p}-{Τ}_{s}\right)A\tau$. Figure 1 displays the illustration of the phenomenon in a geometric format.

The mathematical equations that regulate the trihybrid nanofluid for couple stress fluid are as follows:(1)$$\rho_{Thnf}\frac{\partial v\left(y,\tau \right)}{\partial \tau }=G^{*}+\mu_{Thnf}\frac{\partial^{2}v\left(y,\tau \right)}{\partial y^{2}}-\sigma_{Thnf}B_{0}^{2}v\left(y,\tau \right)-\eta^{*}\frac{\partial^{4}v\left(y,\tau \right)}{\partial y^{4}}+g{\left(\rho \beta_{T}\right)}_{Thnf}\left(Τ\left(y,\tau \right)-{Τ}_{s}\right),$$
(2)$${\left(\rho C_{p}\right)}_{Thnf}\frac{\partial Τ\left(y,\tau \right)}{\partial \tau }=k_{Thnf}\frac{\partial {Τ}^{2}\left(y,\tau \right)}{\partial y^{2}}-\frac{\partial q_{r}}{\partial y}+\mu_{Thnf}{\left(\frac{\partial v\left(y,\tau \right)}{\partial y}\right)}^{2},$$

Here $q_{r}=-\frac{4\omega_{0}}{3\beta_{0}}\frac{\partial T^{4}}{\partial y}$ and $T^{4}\approx \left(4T_{\infty }^{3}\right)T\left(y,\tau \right)-3T_{\infty }^{4}$

The physical IBCs are expressed as follows:(3)$$\left.\begin{array}{l} v\left(y,0\right)=0,\;\;\;\;\;\;\;\;\;\;\;\;\;\;\;\;\;\;\;\;\;\;\;\;\;\;Τ\left(y,0\right)={Τ}_{s},\;\;\\ v\left(0,\tau \right)=0,\;\;\;\;\;\;\;\;\;\;\;\;\;\;\;\;\;\;\;\;\;\;\;\;\;\;Τ\left(0,\tau \right)={Τ}_{s},\;\\ v\left(l,\tau \right)=U_{0}H\left(\tau \right),\;\;\;\;\;\;\;\;\;\;\;\;\;\;\;\;\;\;Τ\left(l,\tau \right)={Τ}_{s}+\left({Τ}_{p}-{Τ}_{s}\right)A\tau ,\\ \frac{\partial^{2}v\left(0,\;t\right)}{\partial y^{2}}\;\;\;\;\;\;\;=\;\;\;\;\;\;\frac{\partial^{2}v\left(l,\;t\right)}{\partial y^{2}}\;\;\;\;\;=\;\;\;\;\;0;\;\;\;t\geq 0.\;\;\;\; \end{array}\right\}.$$

Table 1 presents the mathematical relations between the base fluid and ternary nanofluids, incorporating the effects of different nanoparticle concentrations on the physical properties of the fluid. It outlines the effective thermal conductivity, viscosity, and other relevant parameters for the ternary nanofluid mixture, considering the contributions of alumina, cuprous oxide, and titania nanoparticles in a water-based solution. The relations are derived using established mixing rules and are crucial for predicting the heat transfer and flow behavior of the nanofluid system under varying conditions.

Table 2 provides the numerical values of the nanoparticle concentrations and the base fluid properties used in the study. It details the volume fractions of alumina, cuprous oxide, and titania nanoparticles in the base fluid (water), highlighting their respective contributions to the thermal and rheological behavior of the ternary nanofluid. These values are essential for understanding the impact of varying nanoparticle concentrations on the heat transfer performance and fluid dynamics in the system.

Dimensionless entities include the following:(4)$$\left.\begin{array}{l} \xi =\frac{y}{l},\;u=\frac{v}{U_{0}},\;t=\frac{U_{0}\tau }{l},\;ƛ=\frac{\eta^{*}}{\mu l^{2}},\;P=\frac{l^{2}}{\mu U_{0}}G^{*},\;\Theta =\frac{T-T_{s}}{T_{p}-T_{s}},\; \end{array}\right\}.$$

By incorporating the correlations of the trihybrid nanofluid with the dimensionless entities, Equations (1) and (2) can be expressed as follows:(5)$$\frac{\partial u\left(\xi ,t\right)}{\partial t}=G+ℜ\frac{\partial^{2}u\left(\xi ,t\right)}{\partial \xi^{2}}-Mu\left(\xi ,t\right)-\eta \frac{\partial^{4}u\left(\xi ,t\right)}{\partial \xi^{4}}+Gr\Theta \left(\xi ,t\right),$$
(6)$$\frac{\partial \Theta \left(\xi ,t\right)}{\partial t}={ƛ}_{3}\frac{\partial \Theta^{2}\left(\xi ,t\right)}{\partial \xi^{2}}+E_{c}{\left(\frac{\partial u\left(\xi ,t\right)}{\partial \xi }\right)}^{2},$$ and
(7)$$\left.\begin{array}{l} u\left(\xi ,0\right)=0,\;\;\;\;\;\;\;\;\;\;\Theta \left(\xi ,0\right)=0,\\ u\left(0,t\right)=0,\;\;\;\;\;\;\;\;\;\;\;\Theta \left(0,t\right)=0,\\ u\left(1,t\right)=1,\;\;\;\;\;\;\;\;\;\;\;\Theta \left(1,t\right)=t, \end{array}\right\}.$$

Here,
G=Pμfρfm1U0l,     ℜ=μfm2ρfm1U0l,     M=σfB02lm6ρfm1U0,      Gr=gβΤflm7Τp−Τsm1U02,  η=λμfρfm1U0l,    Pr=μfCpkf,    Re=U0lυ,   Rd=16Τ∞3ω03β0kfm8,   Ec=μfU0m2ρCpfm8Τp−Τsl,    ƛ3=m8m12Pr1+Rd

Here, G,   M,     Gr,    η,    Pr,   Re,   Ec, and $R_{d}$ shows the external pressure gradient, magnetic number, thermal Grashof number, couple stress parameter, Prandtl number, Reynold number, radiation parameter, and Eckert number, respectively.

The following are the constants depending on trihybrid nanofluid correlations.
$$\begin{array}{l} m_{1}=\left(1-\phi_{1}\right)\left[\left(1-\phi_{2}\right)\left\{\left(1-\phi_{3}\right)+\phi_{3}\frac{\rho_{3}}{\rho_{f}}\right\}+\phi_{2}\frac{\rho_{2}}{\rho_{f}}\right]+\phi_{1}\frac{\rho_{1}}{\rho_{f}},\;\;\;\;\\ \;m_{2}=\frac{1}{{\left(1-\phi_{3}\right)}^{2.5}{\left(1-\phi_{2}\right)}^{2.5}{\left(1-\phi_{1}\right)}^{2.5}},m_{3}=\frac{\sigma_{3}\left(1+2\phi_{3}\right)-\phi_{hnf}\left(1-2\phi_{3}\right)}{\sigma_{3}\left(1+2\phi_{3}\right)+\phi_{hnf}\left(1-2\phi_{3}\right)},\\ m_{4}=\frac{\sigma_{2}\left(1+2\phi_{2}\right)-\phi_{nf}\left(1-2\phi_{2}\right)}{\sigma_{2}\left(1+2\phi_{2}\right)+\phi_{nf}\left(1-2\phi_{2}\right)},\;\;\;\;\;\;\;\;\;m_{5}=\frac{\sigma_{1}\left(1+2\phi_{1}\right)-\phi_{f}\left(1-2\phi_{1}\right)}{\sigma_{1}\left(1+2\phi_{1}\right)+\phi_{f}\left(1-2\phi_{1}\right)},\\ m_{6}=m_{3}m_{4}m_{5},\;\;\;\;\;\;m_{7}=\left[\left(1-\phi_{1}\right)\left[\left(1-\phi_{2}\right)\left\{\left(1-\phi_{3}\right)+\phi_{3}\frac{{\left(\rho \beta_{T}\right)}_{3}}{{\left(\rho \beta_{T}\right)}_{f}}\right\}+\phi_{2}\frac{{\left(\rho \beta_{T}\right)}_{2}}{{\left(\rho \beta_{T}\right)}_{f}}\right]+\phi_{1}\frac{{\left(\rho \beta_{T}\right)}_{1}}{{\left(\rho \beta_{T}\right)}_{f}}\right],\\ m_{8}=\left(1-\phi_{1}\right)\left[\left(1-\phi_{2}\right)\left\{{\left(1-\phi_{3}\right)}_{f}+\phi_{3}\frac{{\left(\rho C_{p}\right)}_{3}}{{\left(\rho C_{p}\right)}_{f}}\right\}+\phi_{2}\frac{{\left(\rho C_{p}\right)}_{2}}{{\left(\rho C_{p}\right)}_{f}}\right]+\phi_{1}\frac{{\left(\rho C_{p}\right)}_{1}}{{\left(\rho C_{p}\right)}_{f}},\;\;\;\;\\ m_{12}=m_{9}m_{10}m_{11},\;\;\;\;m_{9}=\frac{k_{3}+2k_{hnf}-2\phi_{3}\left(k_{hnf}-k_{3}\right)}{k_{3}+2k_{hnf}+2\phi_{3}\left(k_{hnf}-k_{3}\right)},\;\;\;\;\;\;\;m_{10}=\frac{k_{2}+2k_{nf}-2\phi_{2}\left(k_{nf}-k_{2}\right)}{k_{2}+2k_{nf}+2\phi_{2}\left(k_{nf}-k_{2}\right)},\\ m_{11}=\frac{k_{1}+2k_{f}-2\phi_{1}\left(k_{f}-k_{1}\right)}{k_{1}+2k_{f}+2\phi_{1}\left(k_{f}-k_{1}\right)}. \end{array}$$

## 3. Fractal–Fractional Model

To generalize the classical model described in Equations (5) and (6), we shall utilize the FFD operator in the Caputo–Fabrizio sense.
(8)Dtα,βFFuξ,t=G+ℜ∂2uξ,t∂ξ2−Muξ,t−η*∂4uξ,t∂ξ4+GrΘξ,t,
(9)Dtα,βFFΘξ,t=ƛ3∂Θ2ξ,t∂ξ2+Ec∂uξ,t∂ξ2,

After applying the fractal derivative operator and corresponding initial condition, we shall get the following:(10)DtαCFuξ,t=βtβ−1G+ℜ∂2uξ,t∂ξ2−Muξ,t−η*∂4uξ,t∂ξ4+GrΘξ,t,
(11)DtαCFΘξ,t=βtβ−1ƛ3∂Θ2ξ,t∂ξ2+Ec∂uξ,t∂ξ2

Here, Dtα,βFF is the fractal–fractional operator [8]:(12)Dtα,βFFft=ωαΓ1−αddtβ∫0tfψexp−α1−αt−ψdψ,     0<α, β≤1.

Equation (12) holds the property: $\omega \left(0\right)=\omega \left(1\right)=1.$

Here, $f\left(t\right)$ is differentiable in the open interval $\left(a,b\right)$, and f is fractally differentiable on (*a*, *b*) with the order β.

## 4. Discretization of the Model

The Crank–Nicolson technique will be used to discretize the model. 

The 1st order FFD is discretized as follows:(13)DταFFfζ,τ=βτβ−1ωα2αfij+1−fijλ+∑j=1mfij+1−m−fij−mλ+Otϖj,m,

Here, $\varpi_{j,m}=erf\left\{\frac{\alpha j}{1-\alpha }\left(m-j\right)\right\}-erf\left\{\frac{\alpha j}{1-\alpha }\left(m-j+1\right)\right\}$.

The Crank–Nicolson scheme provides the discretized form of the first-, second-, and fourth-order derivatives as follows:(14)$$\frac{\partial f\left(\xi ,t\right)}{\partial \xi }=\frac{1}{2}\left(\frac{f_{i+1}^{j+1}-f_{i-1}^{j+1}}{2h}+\frac{f_{i+1}^{j}-f_{i-1}^{j}}{2h}\right)+O\left(t\right),$$
(15)$$\frac{\partial^{2}f\left(\xi ,t\right)}{\partial \xi^{2}}=\frac{1}{2}\left\{\frac{\left(f_{i+1}^{j+1}-2f_{i}^{j+1}+f_{i-1}^{j+1}\right)+\left(f_{i+1}^{j}-2f_{i}^{j}+f_{i-1}^{j}\right)}{h^{2}}\right\}+O\left(t\right),$$
(16)$$\frac{\partial^{4}f\left(\xi ,t\right)}{\partial \xi^{4}}=\frac{1}{2}\left\{\begin{array}{l} \frac{f_{i+2}^{j+1}-4f_{i+1}^{j+1}+6f_{i}^{j+1}-4f_{i-1}^{j+1}+f_{i-2}^{j+1}}{h^{4}}\\ +\frac{f_{i+2}^{j}-4f_{i+1}^{j}+6f_{i}^{j}-4f_{i-1}^{j}+f_{i-2}^{j}}{h^{4}} \end{array}\right\}+O\left(t\right),$$

Given the foregoing discretization of the derivatives, Equations (10) and (11) will have the following form:(17)$$\frac{\omega \left(\alpha \right)}{2\alpha }\left\{\begin{array}{l} \frac{u_{i}^{j+1}-u_{i}^{j}}{\lambda }\\ +\sum_{j=1}^{m}\left(\frac{u_{i}^{j+1-m}-u_{i}^{j-m}}{\lambda }\right) \end{array}\right\}\varpi_{j,m}=\beta t^{\beta -1}\left\{\begin{array}{l} G+ℜ\frac{1}{2}\left(\frac{u_{i+1}^{j+1}-2u_{i}^{j+1}+u_{i-1}^{j+1}}{2h}+\frac{u_{i+1}^{j}-2u_{i}^{j}+u_{i-1}^{j}}{2h}\right)\\ -\eta \frac{1}{2}\left(\begin{array}{l} \frac{u_{i+2}^{j+1}-4u_{i+1}^{j+1}+6u_{i}^{j+1}-4u_{i-1}^{j+1}+u_{i-2}^{j+1}}{h^{4}}\\ +\frac{u_{i+2}^{j}-4u_{i+1}^{j}+6u_{i}^{j}-4u_{i-1}^{j}+u_{i-2}^{j}}{h^{4}} \end{array}\right)\\ -\frac{1}{2}M\left(u_{i}^{j+1}+u_{i}^{j}\right)+\frac{1}{2}Gr\left(\Theta_{i}^{j+1}+\Theta_{i}^{j}\right) \end{array}\right\},$$
(18)$$\frac{\omega \left(\alpha \right)}{2\alpha }\left\{\begin{array}{l} \;\;\;\;\;\;\;\;\;\;\;\;\;\;\;\;\;\frac{\Theta_{i}^{j+1}-\Theta_{i}^{j}}{\lambda }\\ +\sum_{j=1}^{m}\left(\frac{\Theta_{i}^{j+1-m}-\Theta_{i}^{j-m}}{\lambda }\right) \end{array}\right\}\varpi_{j,m}=\beta t^{\beta -1}\left\{\begin{array}{l} {ƛ}_{3}\left\{\begin{array}{l} \frac{\left(\Theta_{i+1}^{j+1}-2\Theta_{i}^{j+1}+\Theta_{i-1}^{j+1}\right)}{2h^{2}}\\ +\frac{\left(\Theta_{i+1}^{j}-2\Theta_{i}^{j}+\Theta_{i-1}^{j}\right)}{2h^{2}} \end{array}\right\}\\ +\frac{Ec}{4}{\left(\frac{u_{i+1}^{j+1}-u_{i-1}^{j+1}}{2h}+\frac{u_{i+1}^{j}-u_{i-1}^{j}}{2h}\right)}^{2} \end{array}\right\}$$

Consider that $y_{i}=ih,$ 0≤i≤M with Mh=1 and $t_{j}=j\lambda$, 0≤j≤N.

## 5. Nusselt Number

Non-dimensional form of the Nusselt number is given by the following.
(19)$$Nu=-\frac{k_{Thnf}}{k_{f}}{\left.\frac{\partial \Theta \left(\xi ,t\right)}{\partial \xi }\right|}_{\xi =0};$$

Figure 2 and Table 3, Table 4 and Table 5, illustrate how heat transfer improves with varying nanoparticle concentrations, also referred to as volume fraction ϕ. For several different kinds of nanofluids, including simple (or unary), hybrid, and ternary hybrid nanofluids, we have computed the Nusselt number. Our investigation focuses on three distinct nanoparticles: alumina, titania, and copper. First, we computed the Nusselt number for each type of nanoparticle separately. Next, we tested hybrid nanofluids made by merging pairs of these nanoparticles. Finally, we compared these findings to those of a ternary hybrid nanofluid, which contains all three types of nanoparticles at once. According to the results, the heat transfer increases for ternary nanofluid *Al_2_O_3_ + TiO_2_ + Cu/H_2_O* up to 3.91% for 1% of nanoparticles. This is followed by various combinations of hybrid nanofluids, such as *Cu/H_2_O* up to 2.80%, *Al_2_O_3_ + Cu/H_2_O* up to 3.44%, and *Al_2_O_3_ + TiO_2_/H_2_O* up to 3.41%, as well as individual simple/unary nanofluids, such as *Cu/H_2_O* up to 2.80%, *Al_2_O_3_/H_2_O* up to 2.35%, and *TiO_2_/H_2_O* up to 2.53%. Similarly, the heat transfer rate of the ternary hybrid nanofluid *Al_2_O_3_ + TiO_2_ + Cu/H_2_O* increases to 12.01% when the quantity of nanoparticles is increased to 4%. The Nusselt number for each pair of hybrid nanofluids was estimated for the same amount of nanoparticles and found to be 10.50% for *Al_2_O_3_ + Cu/H_2_O*, 10.40% for *TiO_2_ + Cu/H_2_O*, and 10.10% for *Al_2_O_3_ + TiO_2_/H_2_O*. Comparably, when individual nanoparticles, or unary nanofluids, are taken into account, the heat transfer increases to 9.51% for *Cu/H_2_O*, 9.42% for *Al_2_O_3_/H_2_O*, and 9.4% for *TiO_2_/H_2_O.* Hence, when comparing ternary hybrid nanofluids, hybrid nanofluids, and unary hybrid nanofluids, it is evident that ternary hybrid nanofluids exhibit the most substantial improvement in heat transfer efficiency. The enhanced thermal conductivity and total heat transfer efficiency of ternary hybrid nanofluids highlight their better effectiveness compared to unary and binary counterparts. This innovation greatly improves the thermal conductivity of pure water, making it more versatile in several fields. By enhancing thermal conductivity, it becomes a highly suitable medium for various applications such as solar collectors, solar water heaters, and solar panels. Moreover, this progress can play a crucial role in enhancing the effectiveness of heat exchangers, cooling systems, and even industrial operations where efficient heat management is of utmost importance. The capacity to enhance thermal efficiency in these various applications underscores the flexible advantages of this novel technique. This enhances the thermal conductivity of pure water, hence increasing the efficiency of solar collectors in converting light energy into various kinds of energy. The quantitative results presented in this study, such as the predicted increase in heat transfer rate, are derived from theoretical models and should be interpreted as model-based predictions rather than experimentally validated values. These predictions rely on the mixing rules specified in Table 1 for properties like density and specific heat, which are based on standard theoretical assumptions. While commonly applied in similar studies, these mixing rules carry inherent limitations that may affect the precision of the numerical results. Accordingly, the values reported here are indicative of general trends rather than exact outcomes. Further experimental validation is necessary to confirm and refine these predictions.

## 6. Results

This work considered the flow of a nonlinear trihybrid couple stress nanofluid in a vertical channel. The classical model is generalized by applying the FFD operator. The solution to the nonlinear fractal–fractional model was obtained numerically. The derived solution is used to create the following figures, which show the effects of several factors, including pair stress parameter, external pressure gradient, Grashof number, Eckert number, fractional order, fractal order, and fractional–fractional order.

The parametric analysis of the fluid flow behavior and its heat transfer against different parameters are portrayed in Figure 3, Figure 4, Figure 5, Figure 6, Figure 7, Figure 8, Figure 9, Figure 10 and Figure 11 where the labels used in the *y*-axis of the figures u(ξ,t) and Θ(ξ,t) represent the velocity and temperature functions in its dimensionless forms. The *x*-axis label, which is ξ, represents the *y*-axis of the channel in its dimensionless form.

Figure 3 illustrates the comparison between fractal, non-integer, and classical order models. The fractal–fractional model is superior in its versatility to both the fractional and the classical model, as it incorporates them by changing their parameters. By setting the parameters accurately, the fractal–fractional model can be simplified and matched with the classical model. Both the non-integer and classical models can be derived from the fractal–fractional model by setting the parameters as α=β=1. This advanced and extended model gives us a range of solutions and integral curves, making them best fitted for aligning the theoretical outcomes with experimental results. Since the fractal–fractional order model includes the fractal dimension, it provides a notable memory effect than other models. The figure also presents a comparison of nanofluid flows with different compositions. The trihybrid nanofluid flow has a consistently better thermal profile compared to the hybrid and unary flows. This can be attributed to the influence of the three distinct nanoparticles. Similar variations in the memory effects have been observed for the thermal field as depicted in Figure 6.

The temperature profiles of the unary, hybrid, and trihybrid nanofluid models are displayed in Figure 4 in relation to the volume fraction ϕ of the various nanoparticles. The temperature distribution behaves in a way that increases with increasing volume fraction ϕ in all of the models. The multifaceted thermophysical characteristics of the distributed nanoparticles are responsible for this observed pattern in the thermal profile. The distributed nanoparticles enhance the thermophysical characteristics of regular water, including its specific heat capacity, density, concentration, viscosity, and more. As a result, the water’s thermal performance improves, and its heat distribution profile exhibits an increasing trend.

Figure 5 shows how the Eckert number affects the thermal field. The Eckert number boosts the thermal field. The pattern in the figure shows that the *Ec* measures the relationship between the kinetic energy of the boundary layer and the enthalpy difference. Heat starts to move from the plate into the fluid when the Eckert number rises. This transfer leads to a higher dissipation of viscosity, resulting in an improvement in the profile of the thermal field.

The effect of the couple stress parameter η on the velocity distribution can be seen in Figure 7. In all the nanofluid models, the velocity profile shows a decreasing trend in its behavior for greater values of the couple stress parameter η. The velocity fields in all the nanofluid (unary, binary, and ternary) models show a decreasing trend for greater values of η. The decreasing variation in the velocity profile is obvious because of the additives (nanoparticles) that are dispersed uniformly in the base fluid (water). When certain additives are dispersed in the fluid, the opposing forces create the couple forces and couple stress in the fluid. Because of these opposing forces in the fluid, the fluid motion retards.

External pressure plays an important role in controlling the flow of fluid in a channel. External pressure can be used to slow down or speed up the flow of fluid. This depends on the direction and strength of the pressure. As the external pressure gradient G impact has also been taken into account in the present phenomenon, Figure 8 has been plotted to check its impact on the velocity field of nanofluid behavior. The figure shows an increasing trend in behavior as the magnitude of the pressure increases. Since the external pressure gradient is related to the normal stresses, the greater the value G is, the greater the pressure on the fluid. Thus, an increasing trend in the velocity field can be observed.

The system of governing equations in the present analysis is coupled with the heat equation due to this, and the momentum (velocity field) equation contains the effect of heat. The effect of heat on the momentum equation is expressed by the thermal Grashof number *Gr*. Figure 9 has been drawn to examine its effect on the velocity field of the nanofluid. Increased Gr values cause the fluid to become more buoyant and less viscous, which accelerates the fluid’s velocity.

In Figure 10, we can observe how the nanofluid flow’s velocity field is affected by the magnetic field parameter *M*. As *M* increases, the profile of the nanofluid velocity shows a decreasing trend. Increasing the magnitude of M produces Lorentz forces in the fluid, and Lorentz forces cause more charges in the fluids. The more electric charges in the fluid, the more resistance will be faced by the fluid to flow in a channel. Therefore, when the magnitude of the magnetic field parameter increases, the decreasing trend in the velocity field is noticed.

In the present model, the flow has been considered in the porous medium. The porous medium has been considered a discontinuous space in geometry; therefore, the fractal derivative operator will help us to handle it better. The porous medium is also important in different practical life situations. Therefore, to check its impact on the velocity field of nanofluid, Figure 11 has been drawn. The porous medium contains small pores in its structure that allows fluid to flow through it. As the value of permeability parameter K increases, the space between interconnected pores increases, which allows the fluid move easily in it. Therefore, an increasing trend in the velocity profile can be observed.

## 7. Concluding Remarks

The objective of this manuscript is to examine the flow of a couple of stress ternary hybrid nanofluids through a vertical channel. During ternary nanofluid formation, three different nanoparticles are combined with a base fluid water. Considering all possible cases, ternary hybrids, binary hybrids, and unary nanofluids, these nanoparticles are considered. The mathematical problem is formulated using partial differential equations (PDEs). The PDE system with integer order is extended using the FFD operator. The Crank–Nicolson numerical algorithm is used to solve the fourth-order nonlinear problem. The main findings of this study can be summarized as follows: the application of the fractal–fractional operator to the integer-order model allows for the exploration of different outcomes.

❖Fractal–fractional operators provide a range of outcomes that closely align with theoretical discoveries and experimental investigations.❖The numerical approach under consideration is capable of analyzing fourth-order nonlinear coupled partial differential equations (PDEs).❖Ternary hybrid nanofluid yields the most favorable outcomes in terms of increasing the heat transfer rate by up to 12.01% when dispersing a 4% concentration of ternary hybrid nanocomposites consisting of *Al_2_O_3_ + TiO_2_ + Cu* in the base fluid, water.❖In binary hybrid nanofluids, the addition of *Al_2_O_3_ + Cu* increases the heat transfer rate of the base fluid water by 10.50%. This is followed by *TiO_2_ + Cu*, which boosts heat transfer by 10.40%, and *Al_2_O_3_ + TiO_2_*, which improves heat transmission by 10.10%.❖When considering the dispersion of nanoparticles separately, *Cu* provides a heat transfer boost of 9.51%, followed by *Al_2_O_3_* with 9.40% and *TiO_2_* with 9.42% heat transfer enhancement.

## Figures and Tables

**Figure 1 nanomaterials-14-01855-f001:**
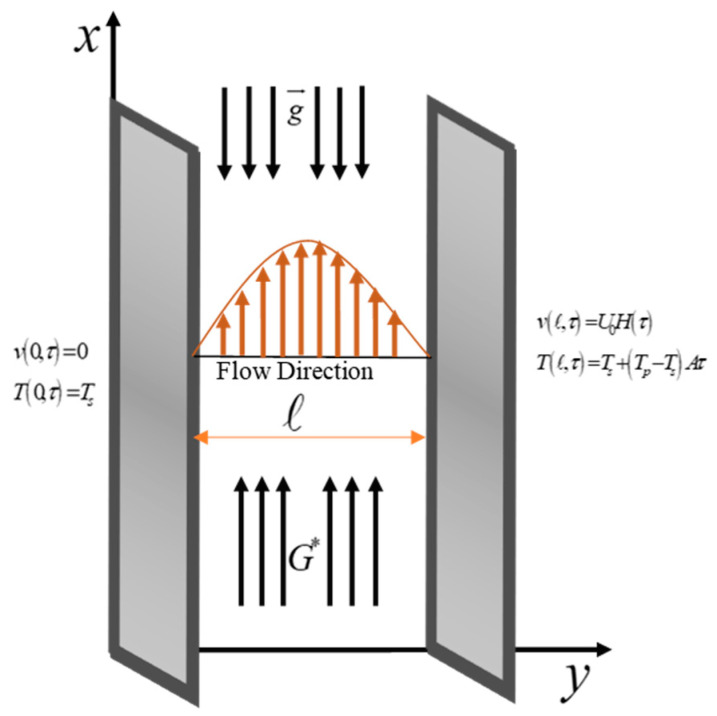
Illustration of the model.

**Figure 2 nanomaterials-14-01855-f002:**
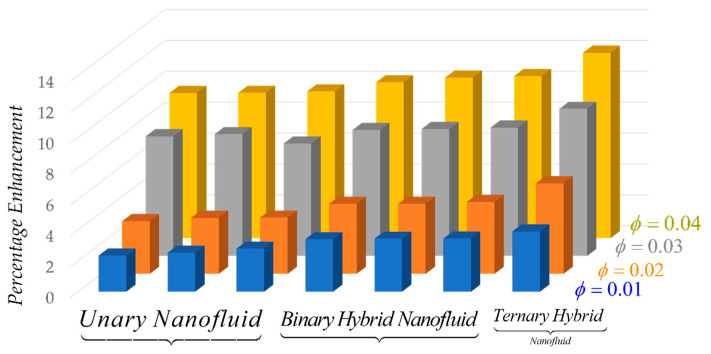
Percentage of heat transfer enhancement of simple/unary, hybrid, and ternary hybrid nanofluids against volume fraction ϕ.

**Figure 3 nanomaterials-14-01855-f003:**
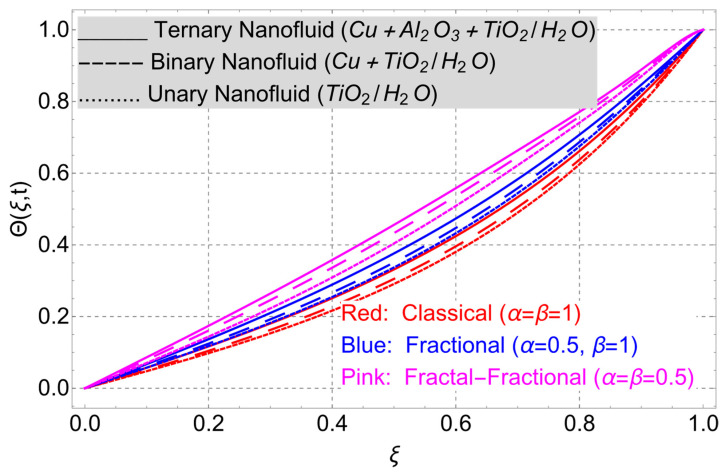
Classical, fractional, and fractal–fractional order temperature profiles when *E_C_* = 2, *ϕ* = 0.02, *G* = 1, *Gr* = 10, *η* = 1, *M* = 2, *t* = 3, and *K* = 1.

**Figure 4 nanomaterials-14-01855-f004:**
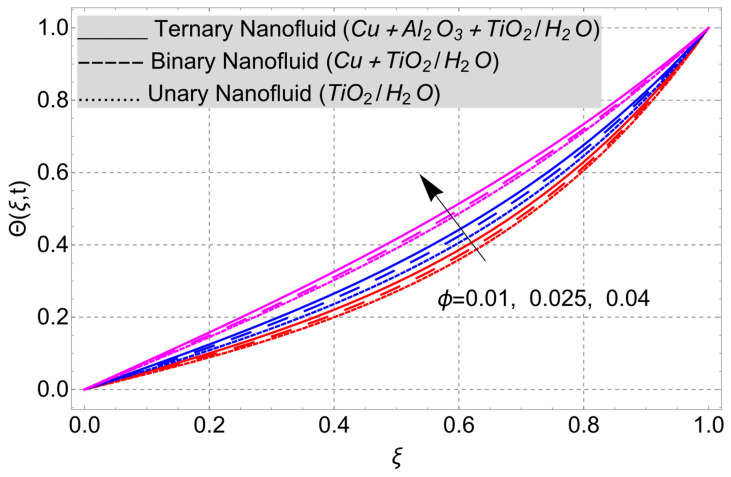
Temperature profile versus volume fraction ϕ when *Ec* = 2, *G* = 1, *Gr* = 10, *η* = 1, *α* = *β* = 1, *M* = 2, *t* = 3, and *K* = 1.

**Figure 5 nanomaterials-14-01855-f005:**
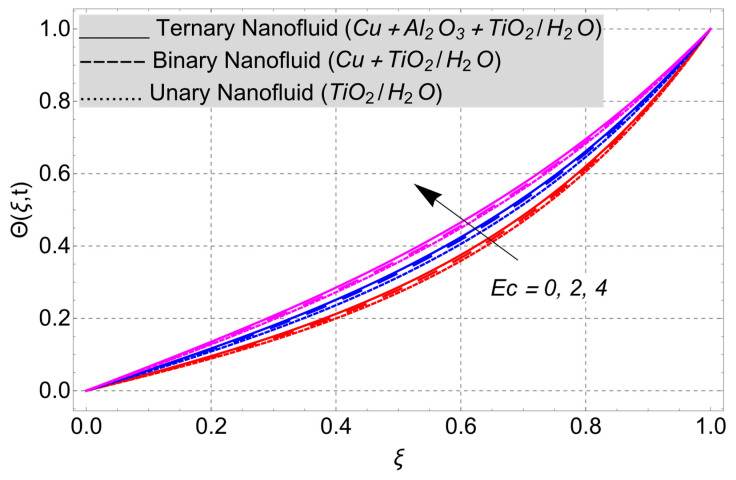
Temperature profile versus Eckert number *Ec* when *G* = 1, *Gr* = 10, *η* = 1, *α* = *β* = 1, *M* = 2, *t* = 3, and *K* = 1.

**Figure 6 nanomaterials-14-01855-f006:**
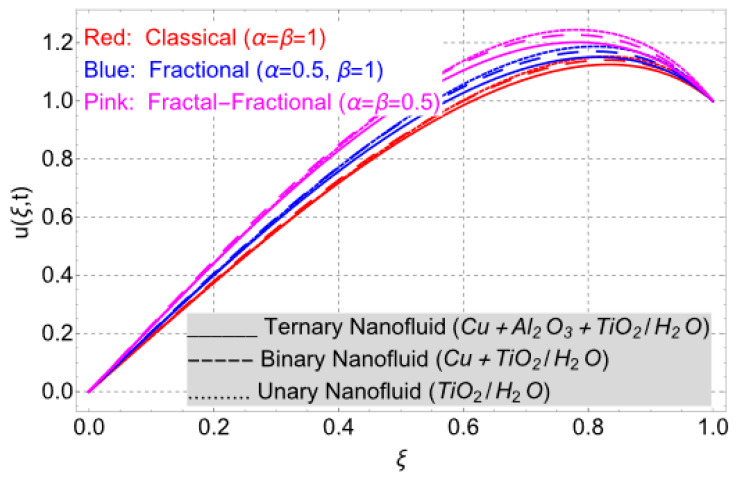
Classical, fractional, and fractal–fractional order velocity profile when *E_C_* = 2, *ϕ* = 0.02, *G* = 1, *Gr* = 10, *η* = 1, *M* = 2, *t* = 3, and *K* = 1.

**Figure 7 nanomaterials-14-01855-f007:**
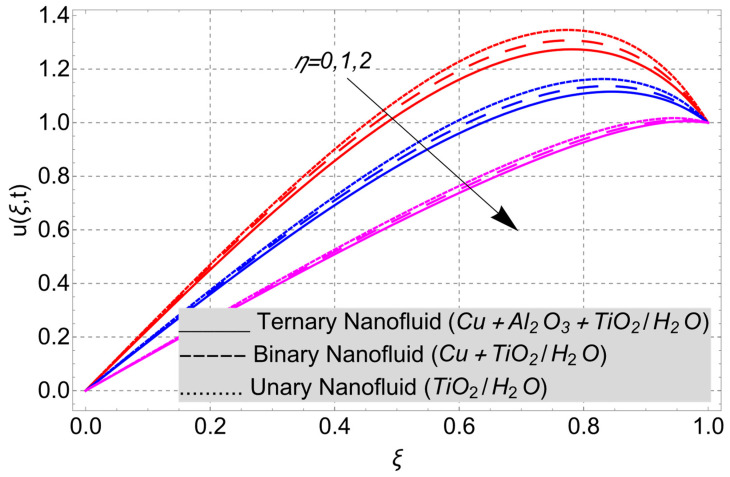
Velocity profile versus couple stress parameter *η* when *E_c_* = 2, *G* = 1, *Gr* = 10, *α* = *β* = 1, *t* = 3, *M* = 2, and *K* = 1.

**Figure 8 nanomaterials-14-01855-f008:**
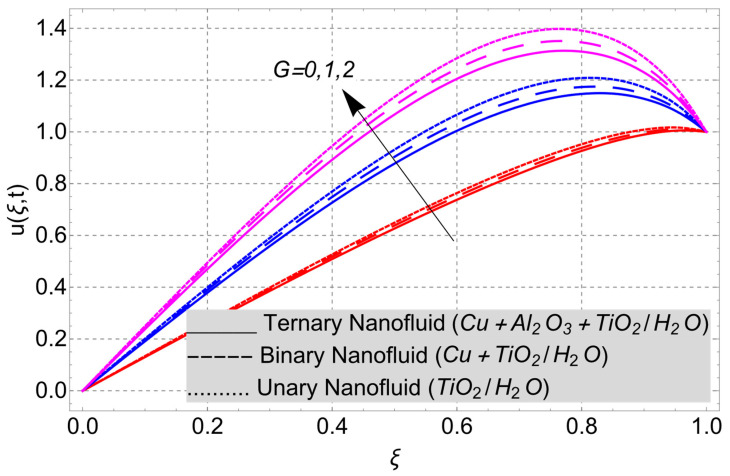
Velocity profile versus external pressure gradient *G* when *E_C_* = 2, *Gr* = 10, *η* = 1, *α* = *β* = 1, *t* = 3, *M* = 2, and *K* = 1.

**Figure 9 nanomaterials-14-01855-f009:**
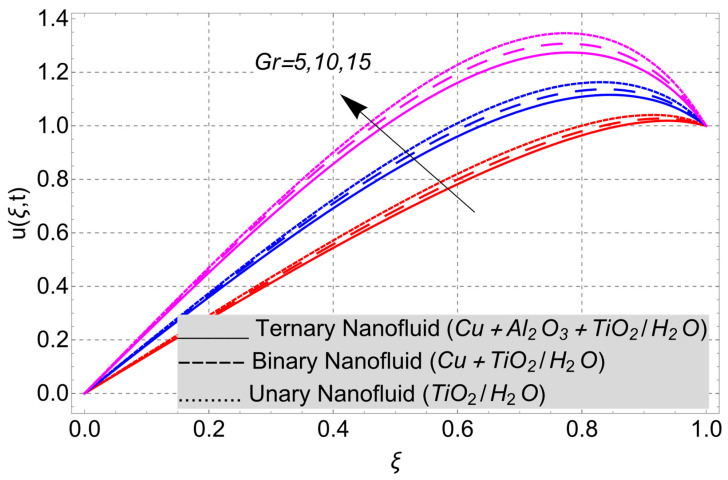
Velocity profile versus thermal Grashof number *Gr* when *E_C_* = 2, *G* = 1, *η* = 1, *α* = *β* = 1, *M* = 2, *t* = 3, and *K* = 1.

**Figure 10 nanomaterials-14-01855-f010:**
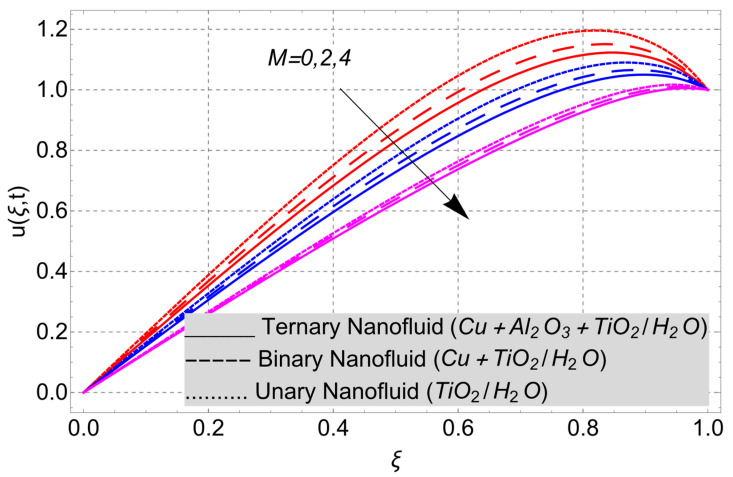
Velocity profile versus magnetic parameter *M* when *E_C_* = 2, *G* = 1, *Gr* = 10, *η* = 1, *α* = *β* = 1, *t* = 3, and *K* = 1.

**Figure 11 nanomaterials-14-01855-f011:**
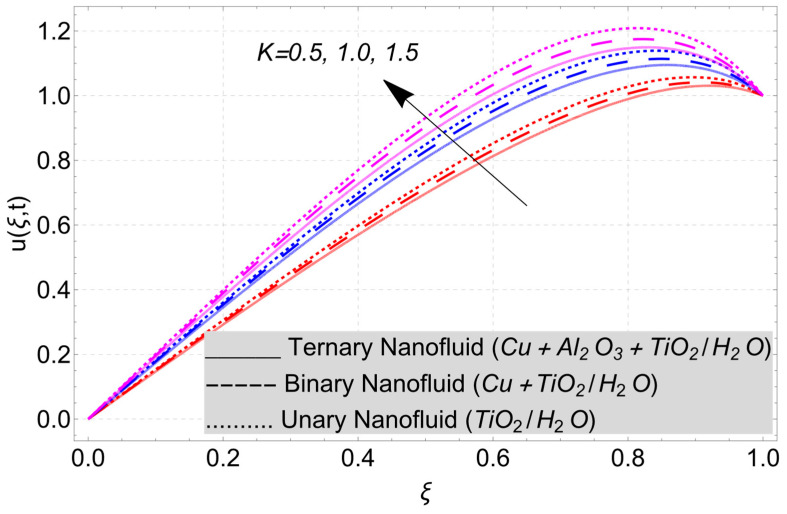
Velocity profile versus porosity parameter *K* when *E_C_* = 2, *G* = 1, *Gr* = 10, *η* = 1, *α* = *β* = 1, and *M* = 2.

**Table 1 nanomaterials-14-01855-t001:** The ternary nanofluid expressions as reported in [26].

Properties	Correlations
Density	$\rho_{Thnf}=\left(1-\phi_{1}\right)\left[\left(1-\phi_{2}\right)\left\{\left(1-\phi_{3}\right)\rho_{f}+\phi_{3}\rho_{3}\right\}+\phi_{2}\rho_{2}\right]+\phi_{1}\rho_{1},$
Viscosity	$\mu_{Thnf}=\frac{\mu_{f}}{{\left(1-\phi_{3}\right)}^{2.5}{\left(1-\phi_{2}\right)}^{2.5}{\left(1-\phi_{1}\right)}^{2.5}},$
Volumetric Expansion	${\left(\rho \beta_{T}\right)}_{Thnf}=\left(1-\phi_{1}\right)\left[\left(1-\phi_{2}\right)\left\{\begin{array}{l} \left(1-\phi_{3}\right){\left(\rho \beta_{T}\right)}_{f}\\ +\phi_{3}{\left(\rho \beta_{T}\right)}_{3} \end{array}\right\}+\phi_{2}{\left(\rho \beta_{T}\right)}_{2}\right]+\phi_{1}{\left(\rho \beta_{T}\right)}_{1}$
Specific Heat Capacity	${\left(\rho C_{p}\right)}_{Thnf}=\left(1-\phi_{1}\right)\left[\left(1-\phi_{2}\right)\left\{\begin{array}{l} \left(1-\phi_{3}\right){\left(\rho C_{p}\right)}_{f}\\ +\phi_{3}{\left(\rho C_{p}\right)}_{3} \end{array}\right\}+\phi_{2}{\left(\rho C_{p}\right)}_{2}\right]+\phi_{1}{\left(\rho C_{p}\right)}_{1}$
Thermal Conductivity	$\begin{array}{l} \frac{k_{Thnf}}{k_{hnf}}=\frac{k_{3}+2k_{hnf}-2\phi_{3}\left(k_{hnf}-k_{3}\right)}{k_{3}+2k_{hnf}+2\phi_{3}\left(k_{hnf}-k_{3}\right)},\;\;\;\;\;\;\;\;\;\;\;\;\;\frac{k_{hnf}}{k_{nf}}=\frac{k_{2}+2k_{nf}-2\phi_{2}\left(k_{nf}-k_{2}\right)}{k_{2}+2k_{nf}+2\phi_{2}\left(k_{nf}-k_{2}\right)},\;\\ \;\;\frac{k_{nf}}{k_{f}}=\frac{k_{1}+2k_{f}-2\phi_{1}\left(k_{f}-k_{1}\right)}{k_{1}+2k_{f}+2\phi_{1}\left(k_{f}-k_{1}\right)},\;\;\;\;\;\; \end{array}$
Electrical Conductivity	$\begin{array}{l} \frac{\sigma_{Thnf}}{\sigma_{hnf}}=\frac{\sigma_{3}\left(1+2\phi_{3}\right)-\phi_{hnf}\left(1-2\phi_{3}\right)}{\sigma_{3}\left(1+2\phi_{3}\right)+\phi_{hnf}\left(1-2\phi_{3}\right)},\;\;\;\;\frac{\sigma_{hnf}}{\sigma_{nf}}=\frac{\sigma_{2}\left(1+2\phi_{2}\right)-\phi_{nf}\left(1-2\phi_{2}\right)}{\sigma_{2}\left(1+2\phi_{2}\right)+\phi_{nf}\left(1-2\phi_{2}\right)},\\ \frac{\sigma_{nf}}{\sigma_{f}}=\frac{\sigma_{1}\left(1+2\phi_{1}\right)-\phi_{f}\left(1-2\phi_{1}\right)}{\sigma_{1}\left(1+2\phi_{1}\right)+\phi_{f}\left(1-2\phi_{1}\right)}, \end{array}$

**Table 2 nanomaterials-14-01855-t002:** Experimental values of base fluid and nanoparticles as reported in [27].

Properties	$\rho \left({\mathrm{kgm}}^{-3}\right)$	$C_{p}\left(J{.\mathrm{kg}}^{-1}k^{-1}\right)$	$k\left({\mathrm{Wm}}^{-1}k^{-1}\right)$	$\beta \times 10^{-5}\left(k^{-1}\right)$
$H_{2}O$	997.1	4.186	0.613	21.00
$Al_{2}O$	3970	765	40	0.85
$TiO_{2}$	4250	686.2	8.9528	0.90
Cu	8933	385	401	1.67

**Table 3 nanomaterials-14-01855-t003:** Percentage enhancement in the heat transfer rate for unary nanofluid.

	Unary Nanofluid					
ϕ	*Nu* for *Al_2_O_3_*	% Enhancement	*Nu* for *TiO_2_*	% Enhancement	*Nu* for *Cu*	% Enhancement
0	3.2801	--	3.2801	--	3.2801	--
0.01	3.3573	2.35	3.3632	2.53	3.3719	2.8
0.02	3.3923	3.42	3.3991	3.63	3.3998	3.65
0.03	3.5345	7.76	3.5392	7.90	3.5192	7.29
0.04	3.5883	9.4	3.5892	9.42	3.5919	9.51

**Table 4 nanomaterials-14-01855-t004:** Percentage enhancement in the heat transfer rate for binary hybrid nanofluid.

	Binary Hybrid Nanofluid					
ϕ	*Nu* for *Al_2_O_3_ + TiO_2_*	% Enhancement	*Nu* for *TiO_2_ + Cu*	% Enhancement	*Nu* for *Al_2_O_3_ + Cu*	% Enhancement
0	3.2801	--	3.2801	--	3.2801	--
0.01	3.3918	3.41	3.3929	3.44	3.3932	3.45
0.02	3.429	4.54	3.4291	4.54	3.4329	4.66
0.03	3.5482	8.17	3.5502	8.23	3.5529	8.32
0.04	3.6129	10.10	3.6198	10.40	3.6239	10.50

**Table 5 nanomaterials-14-01855-t005:** Percentage Enhancement in the heat transfer rate for ternary hybrid nanofluid.

Ternary Hybrid Nanofluid		
ϕ	*Nu* for *Al_2_O_3_ + TiO_2_ + Cu*	% Enhancement
0	3.2801	--
0.01	3.4083	3.41
0.02	3.4726	5.87
0.03	3.5931	89.54
0.04	3.6728	12.01

## Data Availability

The data will be available upon request from the corresponding author.

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
