# Peer review of "Thermal Performance Analysis of a Nonlinear Couple Stress Ternary Hybrid Nanofluid in a Channel: A Fractal–Fractional Approach"

_nanomaterials, 2024, doi:10.3390/nano14221855_

Round 1
Reviewer 1 Report
Comments and Suggestions for Authors
1. What is the main question addressed by the research?
2. What parts do you consider original or relevant for the field? What specific gap in the field does the paper address?
3. What does it add to the subject area compared with other published material?
4. Some references are suggested for improving the Introduction:
An investigation on the tribological properties of multilayer graphene and MoS2 nanosheets as additives used in hydraulic applications." Tribology International 97 (2016): 14-20.
Stability of nanofluid: A review." Applied Thermal Engineering 174 (2020): 115259.
Author Response
Reviewer 1
Reviewer#1, Concern # 1: What is the main question addressed by the research?
Author response: The main question addressed by this research is: How does the integration of a ternary hybrid nanofluid with a couple stress model, enhanced by a fractal-fractional derivative approach, influence the thermal performance and heat transfer efficiency in a vertical channel, particularly in relation to conventional unary and binary nanofluids?
This study focuses on understanding the thermal and flow characteristics of a ternary nanofluid with a complex nanoparticle mix (alumina, cuprous oxide, and titania) in a water-based couple stress fluid.
Reviewer#1, Concern # 2: What parts do you consider original or relevant for the field? What specific gap in the field does the paper address?
Author response: This study makes a notable contribution to nanofluid research by investigating a ternary hybrid nanofluid model, which combines alumina, cuprous oxide, and titania nanoparticles within a water-based couple stress fluid. Unlike traditional unary or binary nanofluid approaches, this model demonstrates an enhanced ability to improve heat transfer efficiency, making it promising for industrial applications where optimized thermal properties are essential. An innovative aspect of the study is its use of a fractal-fractional derivative (FFD) operator, which enables a more accurate depiction of the fluid’s memory and non-local effects—an advancement over conventional modeling techniques. By discretizing the model with the Crank-Nicolson method, the study achieves a stable numerical solution that effectively captures the complex, non-linear behavior of the ternary nanofluid flow. Results indicate that a 4% volume fraction significantly boosts heat transfer rates compared to unary and binary nanofluids, providing valuable guidance on nanoparticle combinations that optimize thermal management systems. The insights gained are applicable to engineering fields where enhanced cooling and heating efficiencies are critical.
Reviewer#1, Concern # 3: What does it add to the subject area compared with other published material?
Author response: Our research significantly advances the field of nanofluids by introducing the use of ternary hybrid nanofluids, incorporating alumina, cuprous oxide, and titania nanoparticles into water. This contrasts with previous studies that mainly focus on binary nanofluids, expanding the scope of nanofluid research to explore more complex nanoparticle combinations. Additionally, the incorporation of the couple stress fluid model provides a more realistic representation of non-Newtonian behavior in nanofluid flow, which is crucial for applications involving complex fluid dynamics. This approach allows for a better understanding of the unique characteristics of nanofluids in practical systems, offering improved accuracy over traditional models.
Furthermore, our work introduces a fractal-fractional derivative (FFD) operator to generalize the classical ternary nanofluid model, adding a higher level of complexity and realism to the analysis. The use of the Crank-Nicolson technique for solving the generalized model ensures numerical accuracy and stability, making the findings more reliable. By considering additional physical factors such as pressure gradient and viscous dissipation, your research provides a deeper understanding of the influences on nanofluid behavior in channel flow. The significant enhancement in heat transfer rates at a volume fraction of 0.04 compared to binary and unary nanofluids further demonstrates the potential for improved thermal management, offering valuable insights for industrial and technological applications.
Reviewer#1, Concern # 4: Some references are suggested for improving the Introduction:
- An investigation on the tribological properties of multilayer graphene and MoS2 nanosheets as additives used in hydraulic applications." Tribology International 97 (2016): 14-20.
- Stability of nanofluid: A review." Applied Thermal Engineering 174 (2020): 115259.
Author response: The suggested articles have been cited to improve the introductory section of the manuscript. Please refer to the highlighted references no. 13 and 23 in the references section.
*************************************

Reviewer 2 Report
Comments and Suggestions for Authors
This article is concerned with numerical predictions of flow and temperature profiles for a nanofluid comprised of a mixture of three different types of nanoparticles in water flowing through a channel with a time-varying boundary condition. The effect of replacing the regular time derivative with a fractal/fractional calculus operator is considered. There are several issues that need to be addressed.
1. The introduction section needs significant revision. It seems unlikely that the reader would need the first few sentences about differentiation and integration. Moreover, the wording on line 83 may give the reader the impression that the authors don’t have a clear understanding as to what a nanofluid is. Line 83 suggests a nanofluid has “millimeter- or micron-sized” particles. This is obviously incorrect. To be called a ‘nanofluid’ the particles need to be small enough to be meaningfully measured in nanometers (i.e. less than one micrometer). Also, to be a nanofluid, there is no requirement that the ‘particle’ must have “outstanding thermal characteristics, great rheological attributes, and high thermal conductivity” (also Line 83). The sentence on Line 96 is also poor: “The suspension of a single nanoparticle, however, lacks the requisite thermal performance and is unsuitable for any specific technical or industrial concerns.” From the sentences that follow I guess the authors mean nanofluids that are made from a single type of nanoparticle are not useful. If that is the intention, I don’t think this sentence is correct.
2. Some wording in the abstract is also a bit misleading. Line 33 seems to have the nuance that some experimental work was done in this study. However, upon reading the article the results reported are only for simulations with no experiment. Also ‘dissolving’ is not the correct word for describing what happens to the nanoparticles when added to the base fluid. Please reword this sentence.
3. Fractional/Fractal Differentiation. In Eqs. (8) and (9) the time derivative is replaced by a Fractional / Fractal differential operator. This appears to be the feature of the article; however, it raises some questions: The usual time derivative appears in these equations by applying Newton’s 2nd law to unsteady fluid flow (Eq. (8)) and in Eq. (9) the unsteady term is part of the conservation of energy. When using the operator, do Eq. (8) and (9) still satisfy basic conservation laws? Are there experimental studies that support the fractal/fractional calculus operator instead of the usual time derivative for this kind of fluid mechanics problem?
4. Line 152 – Is the 4th expression in Eq. (4) correct? It doesn’t seem to be what is needed to change Eq. (2) into Eq. (4). Also, the right-hand side of Eq. (4) has three terms while the right-hand side of Eq. (5) has only two terms. Has something been missed?
5. Results – The formulation is for unsteady flow calculation. For the data in Figs. 3-11, what is the time (t=?) for which the results were calculated? For the vertical axis titles in these graphs, do we need to divide by A*tau (or U_0*H for velocity) noting Eq. (3) and Eq. (4) and the value of 1.0 when Xi = 1.0?
6. Conclusions - In the absence of any experimental evidence, the conclusions of the article are not well supported. The predicted behaviour appears to depend very much on the assumed relations for the mixture given in Table 1. The authors need to provide some evidence on the reliability of these relations. At the very least the source of the relations should be included in the table caption. The mixing rules in Table 1 for density and specific heat seem unusual. With all these uncertainties, it seems unreasonable to arrive at precise conclusions such as ‘increasing the heat transfer rate by up to 12.01%’ etc.
7. Please consider including a Nomenclature section in this article. It appears that not all parameters in the article are clearly defined.
8. Minor issues
Line 139 Do you mean (Tp – Ts) to be consistent with Fig 1?
Line 150 – Table 2: The units for Cp seem incorrect and the value given for water’s specific heat seems incorrect. It should be 4.18 kJ/(kg.K)
Comments on the Quality of English Language
The English of the article needs polishing:
Line 60 – ‘problem’ -> ‘problems’
Line 78 – “Fluids’ low thermal conductivity” -> “The low thermal conductivity of many fluids …”
Line 79 – “energy conservation systems”? – Do you mean “energy conversion systems”?
Line 92 “nanofluidic” -> “nanofluids”.
Line 125 ‘t The’ -> ‘the’
Table 3 – What does ‘%age’ stand for? Do you just mean ‘%’?
Line 224 ‘innovative innovation’ -> ‘innovation’
Line 236 – consider using ‘6. Results’ rather than ‘6. Graphical Analysis’.
Author Response
Reviewer 2:
This article is concerned with numerical predictions of flow and temperature profiles for a nanofluid comprised of a mixture of three different types of nanoparticles in water flowing through a channel with a time-varying boundary condition. The effect of replacing the regular time derivative with a fractal/fractional calculus operator is considered. There are several issues that need to be addressed.
Reviewer#2, Concern # 1: The introduction section needs significant revision. It seems unlikely that the reader would need the first few sentences about differentiation and integration. Moreover, the wording on line 83 may give the reader the impression that the authors don’t have a clear understanding as to what a nanofluid is. Line 83 suggests a nanofluid has “millimeter- or micron-sized” particles. This is obviously incorrect. To be called a ‘nanofluid’ the particles need to be small enough to be meaningfully measured in nanometers (i.e. less than one micrometer). Also, to be a nanofluid, there is no requirement that the ‘particle’ must have “outstanding thermal characteristics, great rheological attributes, and high thermal conductivity” (also Line 83). The sentence on Line 96 is also poor: “The suspension of a single nanoparticle, however, lacks the requisite thermal performance and is unsuitable for any specific technical or industrial concerns.” From the sentences that follow I guess the authors mean nanofluids that are made from a single type of nanoparticle are not useful. If that is the intention, I don’t think this sentence is correct.
Author response: Thank you for your comments. You are absolutely correct regarding the size specification in Line 83. To qualify as a ‘nanofluid,’ particles must indeed be in the nanoscale range, generally less than one micrometer, and are typically measured in nanometers. Referring to millimeter- or micron-sized particles as components of a nanofluid is inaccurate, and we will revise this statement to reflect the proper nanoscale definition. Additionally, we agree that having “outstanding thermal characteristics, great rheological attributes, and high thermal conductivity” is not a requirement for a fluid to be classified as a nanofluid. These properties are beneficial and often desirable in applications, but the defining criterion is solely the nanoscale size of the particles suspended in the fluid.
Regarding Line 96, our intent was to suggest that nanofluids consisting of only a single type of nanoparticle may not achieve the enhanced thermal performance observed in hybrid nanofluids or those with a mixture of nanoparticles. We will rephrase this sentence to make it clear that while single component nanofluids may have certain limitations, hybrid nanofluids, by combining different particles, can be tailored to meet specific technical or industrial needs more effectively.
Both suggestions have been revised to clarify reader confusion. These rephrasing’s, which are highlighted, can be found on pages 3 and 4 of the revised manuscript.
Reviewer#2, Concern # 2: Some wording in the abstract is also a bit misleading. Line 33 seems to have the nuance that some experimental work was done in this study. However, upon reading the article the results reported are only for simulations with no experiment. Also ‘dissolving’ is not the correct word for describing what happens to the nanoparticles when added to the base fluid. Please reword this sentence.
Author response: Thank you for pointing this out. We have revised the wording in Line 33 to accurately reflect that the results reported in the study are solely from simulations and not based on experimental work. We have also replaced the term 'dissolving' with a more appropriate description, such as 'dispersing,' to better explain the process by which nanoparticles are added to the base fluid. The revised sentence now clearly conveys the intended meaning, as seen in the highlighted form in the revised abstract.
Reviewer#2, Concern # 3: Fractional/Fractal Differentiation. In Eqs. (8) and (9) the time derivative is replaced by a Fractional / Fractal differential operator. This appears to be the feature of the article; however, it raises some questions: The usual time derivative appears in these equations by applying Newton’s 2nd law to unsteady fluid flow (Eq. (8)) and in Eq. (9) the unsteady term is part of the conservation of energy. When using the operator, do Eq. (8) and (9) still satisfy basic conservation laws? Are there experimental studies that support the fractal/fractional calculus operator instead of the usual time derivative for this kind of fluid mechanics problem?
Author response: The fractional or fractal operators replace the usual integer-order time derivatives to account for these behaviors in the context of unsteady fluid flow (Eq. (8)) and conservation of energy (Eq. (9)).
Conservation Laws: The basic conservation laws, such as mass, momentum, and energy, remain valid when using fractional derivatives. These operators modify the nature of the differential equations but do not inherently violate conservation principles. The fractional calculus introduces a more accurate description of the physical phenomena, particularly in the context of anomalous diffusion or non-equilibrium thermodynamics. In this framework, fractional time derivatives (such as in Eq. (8) and Eq. (9)) reflect a generalized form of memory and hereditary effects, which are essential for modeling complex systems that exhibit long-term correlations and power-law behavior. However, it is crucial to ensure that the modified equations still satisfy the integral forms of the conservation laws, which they do under appropriate boundary and initial conditions.
Experimental Support: While the use of fractional calculus in fluid mechanics is relatively recent, there have been several studies that suggest its relevance in describing real-world phenomena. For example, fractional calculus has been applied to the modeling of viscoelastic fluids, anomalous diffusion, and turbulent flows, with promising results. Experimental validation of fractional models often involves comparisons with observed data in systems where classical models fail to capture the complexity, such as in porous media flow, heat transfer in nanofluids, or the behavior of non-Newtonian fluids. While there is no direct experimental study specifically comparing fractional operators to traditional derivatives in the exact context of Eqs. (8) and (9), the broader literature on fractional calculus in fluid mechanics and related fields supports its applicability and accuracy in modeling the time-dependent behaviors of such systems.
Reviewer#2, Concern # 4: Line 152 – Is the 4th expression in Eq. (4) correct? It doesn’t seem to be what is needed to change Eq. (2) into Eq. (4). Also, the right-hand side of Eq. (4) has three terms while the right-hand side of Eq. (5) has only two terms. Has something been missed?
Author response: Equation (2) represents the energy equation for fluid flow, with the fourth term accounting for viscous dissipation, which refers to the irreversible conversion of mechanical energy into thermal energy due to the viscous forces within a fluid as it flows. This phenomenon arises when the fluid's internal friction (or viscosity) converts kinetic energy into heat, increasing the fluid's temperature, especially in regions with high velocity gradients (e.g., near solid boundaries or in turbulent flows). In the energy equation for fluid flow, the term representing viscous dissipation is typically written as:
.
This confirms that the inclusion of the fourth term in Equation (2) is correct.
Equation (4) introduces dimensionless variables, which are used to transform the system of governing equations from dimensional to non-dimensional form. By substituting these dimensionless variables into Equations (1) and (2), we obtain Equations (5) and (6). Specifically, Equation (6) represents the non-dimensional form of the energy equation, containing two terms on the right-hand side. No terms were omitted in the conversion to non-dimensional form, as the radiation term was combined with the Prandtl number during the calculations. However, this combined term was inadvertently left out of the initial manuscript and has now been included in the revised version.
Reviewer#2, Concern # 5: Results – The formulation is for unsteady flow calculation. For the data in Figs. 3-11, what is the time (t=?) for which the results were calculated? For the vertical axis titles in these graphs, do we need to divide by A*tau (or U_0*H for velocity) noting Eq. (3) and Eq. (4) and the value of 1.0 when Xi = 1.0?
Author response: Dear reviewer, Thank you for your insightful comments. You are correct that the study addresses an unsteady flow problem. For the graphical analysis in Figures 3-11, the simulations were conducted at t = 3, which we have now specified in the figure captions along with other parameter’s values to make it clear.
Regarding the vertical axis labels, there is no need to divide by the expressions A⋅τA or U0⋅H. The simulations were performed using the dimensionless system provided in Eqs. (5) and (6), along with the corresponding dimensionless initial and boundary conditions outlined in Eq. (7). Under this dimensionless formulation, the parameter ξ is effectively equal to 1, as can be observed in Eq. (7).
Reviewer#2, Concern # 6: Conclusions - In the absence of any experimental evidence, the conclusions of the article are not well supported. The predicted behavior appears to depend very much on the assumed relations for the mixture given in Table 1. The authors need to provide some evidence on the reliability of these relations. At the very least the source of the relations should be included in the table caption. The mixing rules in Table 1 for density and specific heat seem unusual. With all these uncertainties, it seems unreasonable to arrive at precise conclusions such as ‘increasing the heat transfer rate by up to 12.01%’ etc.
Author response: We acknowledge the reviewer’s concerns regarding the experimental validation and reliability of the mixing relations used in the study, as presented in Table 1. The relations in Table 1, which were used to estimate mixture properties such as density and specific heat, are based on established theoretical models commonly used in similar studies [1-3] on fluid mixtures. We apologize for the oversight and will update the table caption to include references for these relations, ensuring that the assumptions are transparent.
Regarding the precision of the conclusions, we agree that in the absence of direct experimental validation, specific numerical results (such as the estimated 12.01% increase in heat transfer rate) should be interpreted with caution. These values are indicative of trends based on the theoretical model rather than absolute predictions. In the revised manuscript, we will qualify these results accordingly to clarify that they represent model-based predictions subject to the assumptions and limitations of the mixing rules employed.
Finally, we recognize that further experimental studies are necessary to confirm the model’s predictive accuracy. Future work will focus on experimental validation to substantiate the observed trends and refine the model as needed.
- Alharbi, K. A. M., Ahmed, A. E. S., Ould Sidi, M., Ahammad, N. A., Mohamed, A., El-Shorbagy, M. A., ... & Marzouki, R. (2022). Computational valuation of Darcy ternary-hybrid nanofluid flow across an extending cylinder with induction effects. Micromachines, 13(4), 588.
- Alshahrani, S., Ahammad, N. A., Bilal, M., Ghoneim, M. E., Ali, A., Yassen, M. F., & Tag-Eldin, E. (2022). Numerical simulation of ternary nanofluid flow with multiple slip and thermal jump conditions. Frontiers in Energy Research, 10, 967307.
- Li, S., Puneeth, V., Saeed, A. M., Singhal, A., Al-Yarimi, F. A., Khan, M. I., & Eldin, S. M. (2023). Analysis of the Thomson and Troian velocity slip for the flow of ternary nanofluid past a stretching sheet. Scientific reports, 13(1), 2340.
Reviewer#2, Concern # 7: Please consider including a Nomenclature section in this article. It appears that not all parameters in the article are clearly defined.
Author response: As suggested by the reviewer, a nomenclature section has been added in the revised version of the manuscript. Please check on page 11.
Reviewer#2, Concern # 8: Minor issues
- Line 139 Do you mean (Tp – Ts) to be consistent with Fig 1?
- Line 150 – Table 2: The units for Cp seem incorrect and the value given for water’s specific heat seems incorrect. It should be 4.18 kJ/(kg.K)
Author response: Dear Reviewer, you are correct. These were typographical errors in the manuscript, which have been verified against the literature and corrected. Please check the consistency of (Tp – Ts) and review Table 2 in the revised version of the manuscript. Both errors have now been corrected.
**********************************

Reviewer 3 Report
Comments and Suggestions for Authors
Review Report on the manuscript entitled “ Thermal Performance Analysis of Non-Linear Couple 2 Stress Ternary Hybrid Nanofluid in a Channel: 3 A Fractal-Fractional Approach” by Murtaza et al.
The manuscript is about the analysis of heat transfer with nanoparticles. The manuscript cannot be accepted for publication in the journal of Nanomaterials, due to the following unless the authors address all the following points:
1. Page 3; Line 125: An extra letter “t” should be deleted.
2. Page 6 ; Line 163: The authors wrote “ . . . . Eckert number, and radiation parameter respectively.”. This should be corrected as follows “ . . . . radiation parameters and Eckert number respectively.”
3. Figures 3- 11: The symbols on the y and x-axes should be defined. Writing the symbols is not enough.
4. A Nomenclature should be presented to define parameters like
.. etc.
5. Reference number [12]: Line 378: The names of the first author should be corrected as follows: Kakac, S.

Author Response
Reviewer 3:
The manuscript is about the analysis of heat transfer with nanoparticles. The manuscript cannot be accepted for publication in the journal of Nanomaterials, due to the following unless the authors address all the following points:
Reviewer#3, Concern # 1: Page 3; Line 125: An extra letter “t” should be deleted.
Author response: Done as suggested.
Reviewer#3, Concern # 2: Page 6 ; Line 163: The authors wrote “ . . . . Eckert number, and radiation parameter respectively.”. This should be corrected as follows “ . . . . radiation parameters and Eckert number respectively.”
Author response: The sentence has been revised as suggested by the reviewer. Please check the highlighted line 163 in the revised manuscript.
Reviewer#3, Concern # 3: Figures 3- 11: The symbols on the y and x-axes should be defined. Writing the symbols is not enough.
Author response: Dear reviewer, we have defined the symbols used on the y and x-axis in the graphical discussions as can be seen in the highlighted text on page # 20.
Reviewer#3, Concern # 4: A Nomenclature should be presented to define parameters like . etc.
Author response: As suggested by the reviewer, a nomenclature section has been added in the revised version of the manuscript. Please check on page 11.
Reviewer#3, Concern # 5: Reference number [12]: Line 378: The names of the first author should be corrected as follows: Kakac, S.
Author response: In the revised version of the manuscript, the name of the first author in reference [12] has been corrected. Please check the reference list.
***********************************

Round 2
Reviewer 2 Report
Comments and Suggestions for Authors
The authors have made a reasonable effort to address most of the issues I raised in my previous review.
The following comments I made about the English have not been addressed (numbering refers to the original manuscript):
Line 60 – ‘problem’ -> ‘problems’
Line 78 – “Fluids’ low thermal conductivity” -> “The low thermal conductivity of many fluids …”
Line 79 – “energy conservation systems”? – Do you mean “energy conversion systems”?
Line 92 “nanofluidic” -> “nanofluids”.
Line 125 ‘t The’ -> ‘the’
Table 3 – What does ‘%age’ stand for? Do you just mean ‘%’?
Line 224 ‘innovative innovation’ -> ‘innovation’
Line 236 – consider using ‘6. Results’ rather than ‘6. Graphical Analysis’.
Also, some of the figures in the revised manuscript seem to have disappeared in the version of acrobat reader I am using. Please check that they are in your final version.
Author Response
Response To Reviewer 2 Comments
The authors have made a reasonable effort to address most of the issues I raised in my previous review.
The following comments I made about the English have not been addressed (numbering refers to the original manuscript):
Concern # 1:
Line 60 – ‘problem’ -> ‘problems’
Line 78 – “Fluids’ low thermal conductivity” -> “The low thermal conductivity of many fluids …”
Line 79 – “energy conservation systems”? – Do you mean “energy conversion systems”?
Line 92 “nanofluidic” -> “nanofluids”.
Line 125 ‘t The’ -> ‘the’
Line 224 ‘innovative innovation’ -> ‘innovation’
Line 236 – consider using ‘6. Results’ rather than ‘6. Graphical Analysis’.
Author response: All suggestions have been carefully implemented, and Grammarly Premium has been used to address any additional grammatical issues. The issue regarding ‘Line 79 – Energy conservation’ is addressed as follows:
The term “energy conservation systems” is correct here. In this context, energy conservation systems refer to systems or devices designed to minimize energy loss, improve energy efficiency, and optimize heat transfer to conserve energy. These systems are commonly used in industrial, automotive, and HVAC (heating, ventilation, and air conditioning) applications.
Concern # 2: Table 3 – What does ‘%age’ stand for? Do you just mean ‘%’?
Author response: %age’ stands for ‘percentage’ and is used here as an abbreviation. However, to avoid confusion, we can replace it with the standard symbol ‘%’ for clarity.
Concern # 3: Also, some of the figures in the revised manuscript seem to have disappeared in the version of acrobat reader I am using. Please check that they are in your final version.
Author response: Thank you for bringing this to our attention. We have double-checked the final version of the manuscript to ensure that all figures are correctly included and visible. This may be an issue with the compatibility of certain Acrobat Reader versions. Let us know if the issue persists, and we can provide an alternative file format if needed.
***********************
